# Recent Progresses on Metal Halide Perovskite-Based Material as Potential Photocatalyst

**Bianca-Maria Bresolin** [1,*] **, Yuri Park** [1] **and Detlef W. Bahnemann** [2,3,4]

1    Separation and Purification Technology, School of Engineering Science, Lappeenranta University of Technology, Sammonkatu 12, 50130 Mikkeli, Finland; Yuri.Park@lut.fi
2    Institut für Technische Chemie, Leibniz Universität Hannover, Callinstr. 3, D-30167 Hannover, Germany; Bahnemann@iftc.uni-hannover.de
3    Laboratory of Nano- and Quantum-Engineering (LNQE), Gottfried Wilhelm Leibniz University Hannover, Schneiderberg 39, 9 D-30167 Hannover, Germany
4    Laboratory Photoactive Nanocomposite Materials, Saint-Petersburg State University, Ulyanovskaya str. 1, Peterhof, Saint-Petersburg 198504, Russia
*    Correspondence: biancabresolin@yahoo.it

**Abstract:** Recent years have witnessed an incredibly high interest in perovskite-based materials. Among this class, metal halide perovskites (MHPs) have attracted a lot of attention due to their easy preparation and excellent opto-electronic properties, showing a remarkably fast development in a few decades, particularly in solar light-driven applications. The high extinction coefficients, the optimal band gaps, the high photoluminescence quantum yields and the long electron–hole diffusion lengths make MHPs promising candidates in several technologies. Currently, the researchers have been focusing their attention on MHPs-based solar cells, light-emitting diodes, photodetectors, lasers, X-ray detectors and luminescent solar concentrators. In our review, we firstly present a brief introduction on the recent discoveries and on the remarkable properties of metal halide perovskites, followed by a summary of some of their more traditional and representative applications. In particular, the core of this work was to examine the recent progresses of MHPs-based materials in photocatalytic applications. We summarize some recent developments of hybrid organic–inorganic and all-inorganic MHPs, recently used as photocatalysts for hydrogen evolution, carbon dioxide reduction, organic contaminant degradation and organic synthesis. Finally, the main limitations and the future potential of this new generation of materials have been discussed.

**Keywords:** metal halide perovskites; renewable energy; photocatalysis; material science

## 1. Introduction

Environmental sustainability and the development of alternative and renewable energy supplies represent two of the major challenges of the current society [1,2]. Rapid urbanization and industrialization have resulted in a remarkable water shortage and a massive contamination of the fresh water available. The increase of human activities, related to hazardous greenhouse gases emissions, has been recognized as the major driver of the present global climate change issues. Moreover, the rapid population growth, coupled with increasing energy demand, are alarming, and require the employment of alternative renewable energy sources for a viable future. Therefore, the development of alternative, renewable and green energies replacing unrenewable and limited fossil fuels, and environmental remediation, have become extremely hot topics in academic research, in order to guarantee a sustainable existence on our planet.

Solar power is a renewable energy source that is sustainable and inexhaustible, unlike fossil fuels. Moreover, it can be considered an attractive alternative due its abundance, easy accessibility

and cleanliness [3,4]. In the recent years, increasing attention has been turned on the applications of solar-light energy, such as solar cells, for heating systems and electricity production. In a year, the solar energy irradiating the surface of the Earth is approximately 3,850,000 exajoules (EJ), and it has been proven that, in the current scenario, only 1% of this energy would meet the energy demand of human beings [5]. Human energy consumption in 2005 was estimated to be 539 EJ globally per year, which is around 0.014% of the total solar energy reaching the Earth [6,7]. Moreover, the solar energy employment can increase the world's safety and human sustainability by decreasing the pollution, lowering the risk of mitigating climate change and reducing such a dangerous dependence on fossil fuels. Solar-light based engineering can provide cost-effective and efficient applications, and recent great developments have been achieved using such a sustainable and green source of power.

From this perspective, photocatalysis is considered one of the main routes for an efficient utilization of solar-light energy. The field of photocatalysis has notably expanded in recent decades, undergoing a rapid development in relation to energetic challenges and environmental issues. In particular, photocatalytic processes have been used for renewable energy production and storage, as well as for the remediation and protection of the environment, in particular for treating polluted air and water [8–10]. The multidisciplinary nature of this technology can be considered among the reasons for the rapid evolution, which counts on the contribution of semiconductor physics, surface sciences, photo and physical chemistry, materials science and chemical engineering. Inspired by natural photosynthesis, photocatalytic processes have been applied in different fields, such as the degradation of organic compounds in the liquid and gaseous phase [11–14], water splitting [15,16], carbon dioxide reduction [17,18] and photocatalytic organic synthesis [19,20]. In general, the photocatalytic process can be defined as the acceleration of reactions in the presence of a photocatalyst, which can provide chemical alteration in another species as a consequence of an initial absorption of light radiation. The process usually refers to heterogeneous photocatalysis, when the two chemical species are in different phases (i.e., liquid–solid). Nowadays, the interest in heterogeneous photocatalysis mainly focuses on semiconductors as photocatalysts, because of their intrinsic optical and electronic properties, including the ability to photogenerate electron–hole pairs [21,22]. Heterogeneous photocatalysis exhibits several advantages over other conventional catalytic processes, such as homogenous photocatalysis, including higher stability and a potential recyclability of the photocatalysts used during the process [23–25].

Titanium dioxide ($TiO_2$) is by far the most used and investigated semiconductor photocatalyst, due to its efficient photostability, non-toxicity and abundance. However, $TiO_2$, with its wide band gap (3.2 eV), limits its applicability in the ultraviolet (UV) wavelengths range, which accounts for only 5% of the total solar-light energy that is potentially available [26]. Moreover, the fast recombination of the photogenerated electron–hole pairs leads to a fast dissipation of the energy that reduces the efficiency in photocatalytic processes [27]. Some traditional approaches to overcoming the mentioned issues include the extending of the absorption edge to the visible light range [28,29], and the decreasing of the recombination rate of the photogenerated electrons and holes [30,31]. In the past, the solutions studied for enhancing the photocatalytic efficiency include the design of suitable band gaps [32,33], the employment of nanosized structures [25,34], the adoption of facet-engineered surface and interface design [35,36], metal and non-metal co-catalysts and dopants utilization [37–40], surface modification [41,42] and the heterostructure or Z-scheme construction [43–46]. However, the low light-absorption ability, the disadvantageous charge carriers' recombination rate, and the low photo-conversion efficiencies of the majority of the photocatalysts, remain among the major challenges to be faced. Table 1 reports some recent reports on frequently employed photocatalysts. However, the development of new materials can open a window to enhancing the current photocatalytic performances.

**Table 1.** Summary of some employed materials for photocatalytic applications.

| Photocatalysts | | | Examples | Photocatalytic Applications | Ref. |
|---|---|---|---|---|---|
| Binary semiconductor photocatalysts | nitrides | $N^{3-}$-based | g-$C_3N_4$ | $NO_x$ oxidation | [47] |
| | oxides | $O^{2-}$-based | ZnO | wastewater treatments | [48] |
| | chalcogenides | sulfides-based | CdS | wastewater treatments | [49] |
| | | selenides-based | CdSe | photocatalytic $CO_2$ reduction | [50] |
| Ternary photocatalysts | ternary oxides | $ABO_3$ | $AgNbO_3$ | hydrogen evolution | [51] |
| | | $AB_2O_4$ | $CaIn_2O_5$ | wastewater treatments | [52] |
| | | $ABO_2$ | $AgGaO_2$ | wastewater treatments | [53] |
| | | $ABO_4$ | $BiVO_4$ | wastewater treatments | [54] |
| | ternary halide | $ABX_3$ | $CsPbBr_3$ | wastewater treatments | [55] |
| | ternary chalcogenides | sulfides-based | $CuGaS_2$ | wastewater treatments | [56] |
| | | selenides-based | $CuGaSe_2$ | hydrogen evolution | [57] |

In the last 20 years, metal halide perovskites (MHPs) have received a remarkable amount of attention as one of the most promising materials in the photovoltaic field, due to their excellent optoelectronic properties. In particular, hybrid organic–inorganic MHPs have often been used in the photovoltaic community in the past 10 years, due to their unprecedented rate of power conversion efficiency [47,48]. A pioneering study, introduced by Miyasaka and their research group, investigated the concept of the self-organization potential of MHPs in the nanoporous TiO$_2$ layer for dye-sensitized cells [49,50]. Since then, hybrid organic–inorganic MHPs have rapidly shown their great potential due to their higher efficiency, in comparison to the more traditional silicon or gallium arsenide-based solar cells [51]. Moreover, the hybrid MHPs have excellent carrier diffusion lengths and superior electronic properties, as well as distinct advantages such as structural simplicity and flexibility [52]. These have been considered as a great advantage in many light-based applications. Despite their remarkable properties, the relatively poor environmental stability of hybrid organic–inorganic MHPs is one of their main drawbacks [53]. All-inorganic MHPs have recently drawn increasing attention, particularly due to their higher stability, but there are further developments needed in order to enhance their efficiency and feasibility.

To the best of our knowledge, there are limited reviews regarding the booming use of MHPs-based materials as photocatalysts; nevertheless, this topic has witnessed incredibly fast development in the last 10 years [53–66] (Figure 1). Herein, we aim to briefly introduce metal halide perovskites, reviewing their optical and electrochemical properties, and especially their potential in photocatalytic applications.

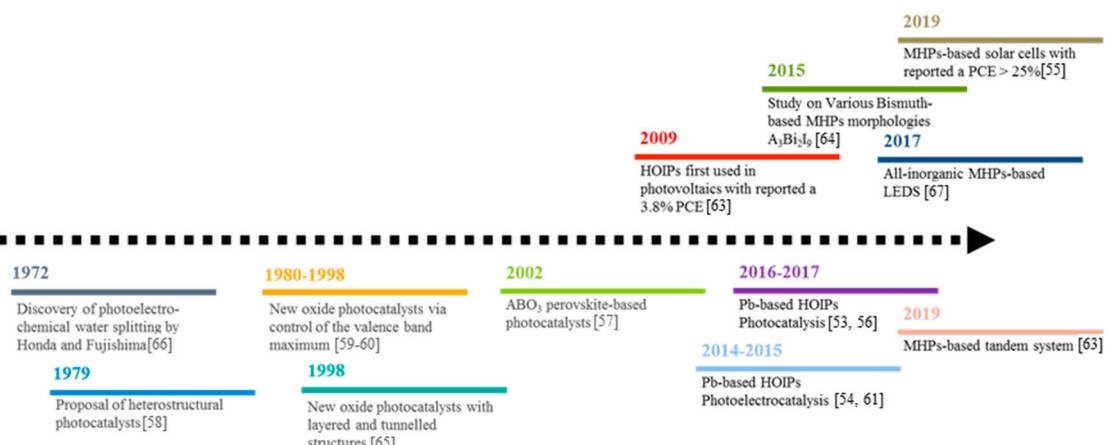

**Figure 1.** Timeline of progress in photocatalysis and metal halide perovskite-based materials.

## 2. Heterogeneous Photocatalysis: Principles

Heterogeneous photocatalysis is an interdisciplinary branch of science that has its source in several areas, including chemistry, physics and biology. Heterogeneous photocatalysis can be considered as one of the most successful approaches used for solar energy harvesting, and many environment-related issues have already arisen at the stage of pilot plants [67–70]. Historically speaking, the first study referred to Fujishima and Honda, who in 1972 reported the photochemical water splitting ability of TiO$_2$ [66].

The photocatalytic process can be related to the ability of a material to absorb photons and promote a reaction, where there is no net storage of chemical energy and the radiant energy is used to carry out a target reaction whose kinetics would be extremely slow, compared to the case of the absence of the photocatalyst. In general, heterogeneous photocatalysis is based on the irradiation of semiconductors by a light source with an energy at least equal to that of its band gap. The process, while varying the details in terms of reactions and mechanisms, can be described in four stages: (I) absorption of light energy to generate electron–hole pairs; (II) separation of the photogenerated charges; (III) transfer of electrons and holes to the surface of photocatalysts; (IV) utilization of photogenerated charges for redox reactions on the surface of the catalyst. The mechanism is displayed in Figure 2A. Figure 2B shows the

reduction and oxidation levels of the principal photocatalytic reactions, with reference to vacuum vs the normal hydrogen electrode (NHE), in terms of the thermodynamic feasibility. The energy of the photoexcited electron must be higher than the corresponding redox level, and the conduction band potential (CB) of the photocatalyst should be located at a higher energy value.

Among the limits of such a mechanism, either on the surface or on the bulk sites of the photocatalysis, the potential electrostatic recombination (III[a]) of the photogenerated charges may dissipate the harvested energy in the form of a radiative and non-radiative recombination, such as light and heat, decreasing the efficiency of the photocatalytic process.

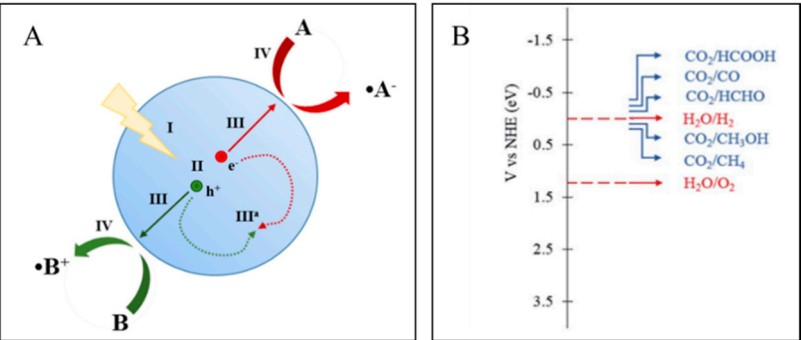

**Figure 2.** (**A**) Steps in photocatalytic reaction process: (I) light absorption, (II) separation of photogenerated charges, (III) transfer of electrons and holes, (III) eventual recombination of charges; (IV) redox reactions where A and B mean chemicals in reductive and oxidative reactions, respectively. (**B**) Energy levels of the important photocatalytic reactions with respect to normal hydrogen electrode NHE at pH = 0.

According to the mentioned principles, there are three main fundamental approaches useful to enhancing the efficiency of a photocatalyst: (i) the increase of its absorption ability in the visible range (45% of the solar spectrum) [71]; (ii) the decrease of the recombination rate of the photogenerated electron–holes pairs [72] and (iii) the evaluation of the redox capacity of the semiconductors according to the target photocatalytic application [73].

## 3. Metal Halide Perovskite

### 3.1. Background on Metal Halide Perovskites

Perovskite materials, with a similar crystal structure of calcium titanate ($CaTiO_3$), were discovered in 1839 by Perovski, a Russian mineralogist [74,75]. Recently, MHPs have been gaining a remarkable amount of attention in the numerous academic fields, yielding over 100 peer-reviewed articles [76] (Figure 3).

MHPs were firstly investigated for their captivating optical properties [77,78], as well as their interesting quantum effects [79–81]. In the past 20 years, MHPs-based materials have witnessed an unbelievable development, especially in photovoltaics, with a remarkable increase of interest also in many other light-driven related applications [76]. MHPs have recently been involved in significant discoveries related to nanometer-sized semiconductors, due to their visible and near-infrared optical abilities, the beneficial effects of their surface defects and their specific electronic structure [48,50,82,83]. Solar-cells have shown to be a great implementation of MHPs due to their easy preparation and remarkable efficiencies, leading to results comparable to the more traditional semiconductors-based devices, such GaAs and Si, due to their high crystallinity and low temperature processing [84]. The evaluation of the potential of MHPs as sensitizers on mesoporous $TiO_2$ electrodes was started in 2005 in the Miyasaka group [85], inspired by the quantum photochemistry research of Teshima and Kojima [86,87]. Preliminary studies with methylammonium lead halide perovskite nanocrystals ($CH_3NH_3PbI_3$) showed promising results in visible light absorption ability when deposited on

a $TiO_2$ electrode. Schmidt et al. in 2014 published the first solution-based colloidal approaches based on lead halide hybrid organic–inorganic MHPs [88]. Recently, remarkable advances in the exploration and development of all-inorganic MHPs have also been achieved. Protesescu et al. [89] investigated the ability of different cesium-based MHPs to act as entire visible spectrum-absorbers, with surprising and remarkable quantum yield efficiencies. Nowadays, the discoveries associated with the remarkable optical and electrical properties of MHPs have extended to several other applications, such as light-emitting diodes, photodetectors, X-ray detectors, memory devices, and many others [90]. The interdisciplinary interests of MHPs, necessitating expertise in chemistry, physics and optoelectronics, may challenge a great part of the research community, leading to intensive and practical developments in the near future.

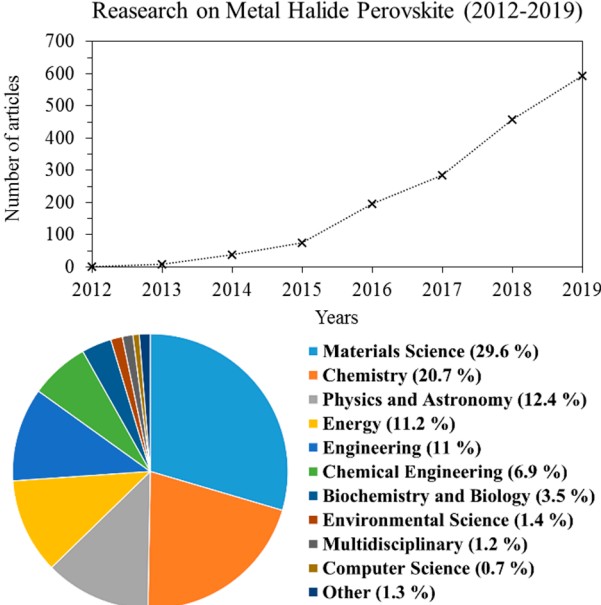

**Figure 3.** Trends of the published scientific articles on metal halide perovskites and academic trends. Analysis search results by Scopus.com.

### *3.2. Perovskites' Fundamental Structure, Composition and Applications*

MHPs have an $ABX_3$ structure (Figure 4), whcih contains a hybrid organic–inorganic or inorganic monovalent cation in the A-site position (methylammonium $MA^+$, formamidinium $FA^+$, cesium $Cs^+$ or mixed cations), a divalent oxidation state IVA metal element in the B-site ($Pb^{2+}$, $Sn^{2+}$, $Ge^{2+}$ ... ), and a halide anions in the X-site ($I^-$, $Br^-$ or $Cl^-$).

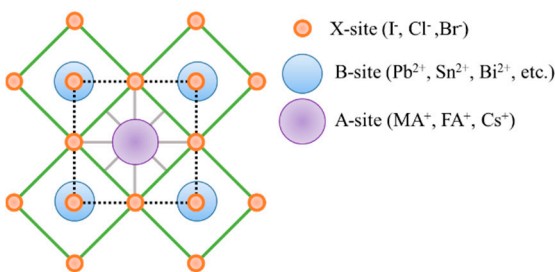

**Figure 4.** General crystal structure of metal halide perovskites.

When the A-site is occupied by organic cations ($MA^+$ or $FA^+$), we usually refer to these as hybrid organic–inorganic MHPs. In recent research, in order to overcome the relative environmental instability of hybrid organic–inorganic MHPs, scientists have efficiently replaced the organic cations with inorganic

metals (all-inorganic MHPs) without decreasing the electro-chemical properties [91]. Moreover, intensive research into structural and compositional modifications have produced breakthroughs in MHPs efficiencies [92–94].

Generally speaking, the cation selected for the A-site is responsible in the determination of the orientation cage within the structure, and its manipulation may lead to the beneficial transition of the optoelectronic properties [95–97]. The size of the A-cation adversely affects the symmetry of the structure, causing distortions on the B–X bond which controls the electronic properties in the perovskite framework [98]. Among the organic cations, methylammonium (MA$^+$) has been widely used because of its radius (1.8 Å), resulting in lower packing symmetry, higher band gap and higher efficiency, and it appears to be the more suitable in solar cells [95]. In general, inorganic elements exhibit better stability compared to organics; therefore, the organic cations have been beneficially substituted with cesium cations (Cs$^+$) in what have become known as all-inorganic MHPs [99].

The B-site is generally occupied by elements belong to the IVA group of the periodic table. Its modulation was found to be affected by proceeding along the periodic table group, with a reduction in the stability of the divalent oxidation state, decreasing as a consequence of the band gap and electron pair effect, due to the enhancement of its electronegativity and covalent characters [72,100]. Despite lead-based MHPs gaining a lot of success, partial or total substitution of lead (Pb) with other ions, such as tin (Sn), may led to increases of its absorption ability and environment stability [101]. Moreover, ions doping, in particular with environmentally friendly ions (e.g., Ca, Mn, Sr, Bi), has been recently considered, in order to find an equilibrium between good performances and environmental aspects [102–107], leading, in particular, to an increased stability.

Finally, the X-site is occupied by a halogen element that has a great influence on the valence band energy of the MHPs, according to its electronic features. In particular, the valence bands and the associated electron binding energies of MHPs have been recorded to decrease after applying the XVII group elements, going from Cl to I. Manipulation with mixed halogen ions has led to an enhancement in the thermal stability, in comparison to the single halogen ion-based MHPs counterparts. Mixed Cl and I halide perovskite have shown improved efficiency in the pioneering work of Lee and co-workers [50]. Previous researchers have further suggested that mixed Cl and I halogens may be beneficial for the growth of crystal domains, due to a slower crystal lattice generation process, leading to a decrease in the grain boundary number, enabling the charge carriers to move across the structure, thus reducing charge recombination [108]. The introduction of Br-halide ions resulted in an increase in the quality of the final material in terms of the symmetry of the lattice, its stability under a humid environment, and its morphology and optical properties [109,110]. However, total substitutions of Br resulted in photoinduced phase segregation, and thus the mentioned approach still need further insights [111].

In general, several compositional factors are not entirely understood, including long-term environmental stability, toxicity and final efficiencies, and these are the main points to be addressed in the future in order to assure the feasibility of MHPs.

MHPs attract our attention mainly for the rapid development they have recently achieved in the photovoltaic field, by accomplishing notable progresses in only a few decades. Among the traditional applications where MHPs have witnessed extraordinary developments, one may cite solar cells and light emitting diodes.

Research into perovskite-based solar cells started in 2005, with the aim of examining the possibility of using halide perovskites as sensitizers on mesoporous TiO$_2$ electrodes. The halide perovskites were firstly applied in the dye-sensitized technology in solar cell devices [112]. Lead-based hybrid MHPs (CH$_3$NH$_3$PbI$_3$) adsorbed on TiO$_2$ film was firstly discovered as a potential alternative to the traditional solar cell materials, showing a remarkable photocurrent efficiency [113]. Moreover, if any significant differences were observed, the tests were performed on a different substrate, verifying the interesting ability of the MHPs [51]. In addition, the researchers noticed that the transfer of photogenerated charge carriers could directly occur on the perovskite, resulting in the non-necessity of an electron-accepting layer. Thus, perovskite solar cells may be fabricated from direct junctions between the perovskite,

the n-type TiO$_2$ and p-type hole-transport material (HTM) films. Following this concept, a planar pin junction was prepared by co-evaporation of CH$_3$NH$_3$I and PbCl$_2$, and this structure further increased the photocurrent efficiency [114]. As reported by McMeekin et al. [115], by employing a tandem architecture, halide perovskite-based photovoltaic cells can potentially boost the efficiency of the more commercial modules based on silicon.

On the other hand, perovskite light-emitting diodes (LEDs) are gaining considerable attention for applications in next-generation displays and lighting. Tremendous efforts have been devoted to implement all-inorganic MHPs in this field [66,116], leading to remarkable achievements [117,118]. In summary, the great performance of MHPs-based LEDs can be reduced to three fundamentals: (i) a balanced injection of charges, (ii) a minimal non-radiative charges recombination loss, and (iii) efficient extraction of the emitted photons [119,120]. New strategies have been recently applied in order to further enhance the efficiency of MHPs-based LEDs. Among them, the transition from 3D bulk to 2D layered structures has resulted in an increase in radiative recombination and efficient energy funneling, leading to nearly 100% internal quantum efficiency [121]. Nevertheless, the improvement in device stability still remains among the major challenges, and it has been previously reported that oppression of ion migration, with additives or blocking layers, may further optimize the MHPs' interface connection [117].

### 3.3. Outstanding Properties of MHPs as Photocatalyst

MHPs have mainly emerged for their unique optoelectronic properties, including favorable band gaps for extended light absorption in the visible light range, high charge carrier mobility, and long charge diffusion lengths for a reduced recombination rate of photogenerated electron–hole pairs. These properties have made MHPs remarkably suitable for photocatalytic applications, as remarked in many recent reports [5,19,82,84,93,99,122]. In particular, the important factors that support the potentially superior photocatalytic performance of MHPs include the following: (1) a high optical absorption coefficient with easily tunable band gaps; (2) a long carrier diffusion length and suppressed recombination rate that enhance the charge carriers lifetime; and (3) a well-balanced charge transfer suitable for redox reactions [85].

### 3.3.1. Extended Optical Absorption Range

The light absorption ability of the MHPs was found to be enhanced when compared with traditional semiconductors (e.g., TiO$_2$), contributing to the increase of photon-to-carrier conversion efficiency, as demonstrated in many perovskites-based solar cells [123]. A suitable band gap is fundamentally required in order to maximize the optical absorption of photocatalytic materials. In theory, the range of visible light absorption becomes wider, and the efficiency of light utilization can be increased, by narrowing the band gap values [122]. MHPs-based nanostructures have recently emerged as low-dimensional semiconductors of great interest in photovoltaics, photonics and optoelectronics, and extensive efforts have been made towards their controlled synthesis. Among their unique properties, notable are their exceptionally good light absorption in the visible region and their tunable photoluminescence, which have witnessed increasing attention in different applications and technologies [124].

The engineering of the band structures is considered fundamental in the rational tuning of the electronic and optical properties of perovskite nanostructures, and it is among the key means of achieving multifunctional optoelectronic efficiency [125]. In general, the MHPs have a direct band gap ranging across the entire visible spectrum, meaning that, in a photocatalytic application, the charge carriers can be efficiently generated at lower energies [125,126]. Among the advanced properties of MHPs has been reported an easy approach to achieving seamless tuning of the band gap, by simply changing the mixing ratio of the halide elements (e.g., I–Br, Br–Cl) [127]. Protesescu et al. showed that the emission wavelength of CsPbX$_3$ (with X = Cl, Br or I) can be shifted from 410 nm to 512 nm, and then up to 685 nm, changing only the halogen element in the X-position to Cl, Br or I, respectively.



In the same research, mixed halogen composition led to the potential to absorb light through the entire range of the solar spectrum [128]. The modification of the band gaps of MHPs is very interesting with regards to targeted device design that requires specific energy levels. In recent years, it has been reported that the efficiency of MHPs-based devices can be compared to the state-of-the-art of quantum dots and organic materials as traditional solar-light absorbers [117,129,130].

The nature of the electronic structure of MHPs, and the formation of energetic bands through the hybridization of metal and halogen orbitals, has led to successful results in photocatalytic applications [131]. In contrast with the conventional semiconductors, whose band gaps are formed between bonding and antibonding orbitals, in MHPs the valence band edge shifts in energetic levels, according to the halogen component, while only small changes are ascribed to the conduction band edges [132]. In this case, the A-site cation produces the major effect on the band gap value of the perovskites. It was reported that, with shifting from inorganic cesium ($Cs^+$) to organic $MA^+$ and $FA^+$ cations, as the sizes of cations respectively increase, the band gap of the corresponding material red shifts. This behavior was ascribed to an enhancement in the tilting angle of metal–halogen bonds and the distortion of the crystal structure [133,134]. The B-site's composition may also influence the optical properties, most likely due to the electronegativity of the metal element employed [72,100]. It can be concluded that the final band gap, and the connected optoelectronic properties of the MHPs, strictly depend on the final stoichiometry, if comparable sizes and shapes are maintained. On the other hand, the Stokes shift increases as the size of the nanoparticles is reduced, due to the formation of a confined holes state delocalized across the entire nanocrystal, showing the size-dependency of MHPs' properties [133].

In terms of quantum yield (QY), the electronic structure and the band gap formed between the antibonding orbitals lead to the predominant formation of shallow traps, allowing typically remarkably high values, and a strong defect tolerance. Higher values of QY were reported for inorganic Cs-based and hybrid MA- and FA-based halide perovskites, whereas Cl-based MHPs have much lower values of QY with respect to their Br and I counterparts, due to the small size of the Cl-halogen anions affecting the defects in the crystal structure [131]. Furthermore, the metal-ions doping approach has exhibited the potential enhancement of the optical properties of the halide perovskite. For example, it was reported that when Pb was partially replaced by other cations, such as Mn, the structure may lead to a strong Stokes-shifted emission, resulting in a higher utilization of solar energy [135]. On the contrary, blue shift in the band edge was detected by doping the halide perovskite with other cations ($Sn^{2+}$, $Cd^{2+}$, $Al^{2+}$ and $Zn^{2+}$). These effects can be ascribed to the contraction of the original MHPs' lattice, which may result in wide band gaps and lower absorption abilities [136,137]. Thus, dopant atoms led to a potential modification of the absorption wavelengths, as was demonstrated in the literature, with non-obvious beneficial effects [138].

The morphology of MHPs recently gained more attention with regards to the design of MHPs-based materials, in particular transferring from bulk to low dimensions (0D, 1D, 2D and 3D) nanomaterials. The nanostructures have shown distinctive optoelectronic properties in comparison with their bulk counterparts, due to their quantum-confined effects. The tuning of the band structures of the perovskite nanostructures allows the design of novel, efficient and functional light-active MHPs, leading to a new degree of freedom in the modification of their optoelectronic characteristics. Previous studies have shown that a band gap between 1.34 and 1.5 eV can effectively absorb visible light, but at lower values, the photogenerated charges recombination rate negatively increases. According to this assumption, the energy band diagram of 3D perovskite materials seems to be the most suitable to satisfy the mentioned requirements [139]. On the other hand, when the dimension was reduced to 2D or 1D, the band gap appeared to enlarge, which can be beneficial for some solar light-driven applications [124]. Thus, the control of the compositions, structures and dimensions can affect the band gap of MHPs, leading to a wide tunability with regards to light emission, and absorption ranging from ultraviolet to the near-infrared wavelength [125].

### 3.3.2. Direct Generation of Free Charge Carriers

Exploring charge carriers kinetic is essential to the design of new photocatalysts, because it allows us to reveal the photocatalytic mechanisms and to optimize the material efficiency.

The semiconductor-based photocatalysis is initiated by the absorption of a photon with an energy at least greater than the band gap energy of the photocatalyst, inducing inter-band transition, with the formation of electrons and holes in the conduction and valence bands, respectively. The electron–hole pairs are photogenerated within a femtosecond time, thus they tend to easily recombine in the bulk or on the surface of the photocatalyst, with the subsequent loss of energy in terms of light or heat [140]. Then, the non-recombined charge carriers can migrate to the surface, and they can be trapped before undergoing the interfacial charge transfer. Afterwards, the redox reactions can occur through interfacial charge transfer, influenced by the energetic position of the trapped states. Thus, the overall photocatalytic efficiency is dependent firstly on the pico/nanosecond competition between charge carrier recombination and trapping, and secondly on the micro/millisecond recombination of the trapped charge carriers and interfaces [71]. Thus, energy level alignments at the interfaces, in strict relation with the band gap, play a key role in the engineering of the charge transport behavior of the semionductors.

In the MHPs-based material, increases in the separation and extraction of photo-generated carriers have been recorded as coupled with a decrease in the recombination of charge carriers. Therefore, in the majority of the solar-light driven applications, the MHPs can act both as a charge generation and a transport medium [122]. As for their other properties, the performance of MHPs greatly varies depending on several factors, including crystal structure, nanoscale morphology, microstructure, and the hierarchical architecture of the final [128,141]. However, the easy control over the band alignment in MHPs represents a great advantage, and helps us provide novel and functional materials for different light-driven applications.

### 3.3.3. Long-Range Balanced and Ambipolar Charge Carrier Transport Properties

The diffusion length of the photogenerated carrier has been considered one of the strongest gaps for screening materials for light-driven processes [142].

In the field of heterogeneous photocatalysis, carrier diffusion and transport are important parameters to be consider. The electronic structure, including determination of the absorption coefficient and the charge carrier concentration, remarkably influence carrier lifetime and diffusion length. After the photogeneration of the electrons and holes, excited carriers subsequently undergo separation and transport from the bulk to the surface of the semiconductor. In photocatalytic reactions, charge separation and transport processes are a primary concern [143]. These phenomena, comparable to an electron flow, such as current, are mainly influenced by parameters such as the diffusion coefficient, and the gradient, mobility and concentration of the electrons and holes. As the electronic structure predominantly determines the effectiveness of good mobility, a long carrier diffusion length represents a key advantage as regards the intrinsic properties of a semiconductor [144].

Scientific breakthroughs with MHPs in 2013 were focused on the potential use of MHPs as promising light-harvesters, along with their beneficial electronic properties, such as a long photogenerated diffusion and lifetime, which can be compared to the traditional semiconductors [142,145,146]. Studies on photophysical processes, such as transient absorption spectroscopy, have recently confirmed the effectiveness of charge carrier dynamics in monitoring the photogenerated carriers, in terms of photon absorption, vibrational relaxation and exciton generation and separation [147–149]. The advantageous properties include balanced long electro-hole diffusion lengths that surpassed those of typically used photovoltaic materials [145], high charge carrier mobility and lifetime [150], and beneficial interfacial charge transfer dynamics, as well as still-uncovered electron injection dynamics [151]. Moreover, MHPs have been shown to possess an ambipolar charge carrier transport capacity, which can assure potentially outstanding photocatalytic properties, with balanced hole and electron mobility under light illumination. Recent studies have revealed that the ambipolar

performance can be considered an intrinsic property MHPs as a feature independent from the device architecture [152].

### 3.3.4. Point-Defect Tolerance and Reduced Recombination Centers

The presence of point-defects on a photocatalyst can affect its structural, electronic and optical properties, and can thus be used as a strategy for improving its performance [153].

As for other semiconductors, the intrinsic optoelectronic features of MHPs are greatly determined by several factors, including the presence of beneficial defects in the crystal structure, which, in the case of this family of materials, can be easily controlled by synthesis [154]. Both experimental and theoretical studies have recently demonstrated that defects, including vacancy, impurity and the presence of interstitial atoms, may lead to decreases in band gap value, enhancing the light absorption through the visible range and resulting in higher photocatalytic activity [155,156].

It seems very useful for the future to gain deeper insights into the potential defect science of MHPs, which seems to be essential for the enhancement of the efficiency of materials in light-driven technologies.

## 4. Photocatalytic Applications

### 4.1. Photocatalytic $CO_2$ Reduction

$CO_2$ conversion to fuels has been promoted as a climate change mitigation measure, able to reduce the use of fossil fuels with a simultaneous beneficial recycling of carbon [157,158]. Thus, addressing the problems of the depletion of fossil fuels and global warming appears to be an urgent task, and the search for valid approaches to maintaining the atmospheric $CO_2$ level has become fundamental and urgent [159,160]. Artificial-based photocatalytic processes, supported by globally available and inexhaustible solar-light energy, have attracted particular attention as regards the conversion of $CO_2$ to solar fuels [161]. In this scenario, the remarkable optical and electrochemical properties of MHPs may represent a pivotal advantage for their success as high-efficiency solar energy-based photocatalysts [162]. Recently, different MHPs have been investigated as potential photocatalysts for the reduction of $CO_2$, demonstrating promising results (Table 2).

**Table 2.** Comparison of $CO_2$ reduction efficiency of different metal halide perovskites-based photocatalysts reported in the literature.

| Photocatalyst | Irradiation | Medium | Averaged Electron Yield μmol g$^{-1}$ | Selectivity % | CO μmol g$^{-1}$ | CH4 μmol g$^{-1}$ | H2 μmol g$^{-1}$ | Ref. |
|---|---|---|---|---|---|---|---|---|
| CsPbBr$_3$ QDs 8.5 nm | 300 W Xe, AM 1.5G filter | H$_2$O/ethyl acetate | 20.90 | 99 | 34.10 | 12.20 | 0.80 | [163] |
| CsPbBr$_3$ QDs 11.6 nm | 300 W Xe, AM 1.5G filter | H$_2$O/ethyl acetate | - | - | 24 | 9 | 0.60 | [163] |
| CsPbBr$_3$ QDs 3.8 nm | 300 W Xe, AM 1.5G filter | H$_2$O/ethyl acetate | - | - | 18.10 | 6.90 | 0.40 | [163] |
| Cs$_2$AgBiBr$_6$ unwashed | 300 W Xe, AM 1.5G filter | ethyl acetate | 16.20 | 100 | 5.50 | 0.65 | - | [164] |
| Cs$_2$AgBiBr$_6$ in Abs EtOH | 300 W Xe, AM 1.5G filter | ethyl acetate | 105.30 | 100 | 14.10 | 9.60 | - | [164] |
| Bulk Cs$_2$AgBiBr$_6$ in Abs EtOH | 300 W Xe, AM 1.5G filter | ethyl acetate | 5.60 | | - | - | - | [164] |
| CsPbBr$_3$ QDs | 100 W Xe, AM 1.5G filter | ethyl acetate | 284.70 | 99.3 | 49.50 | 22.9 | 1.07 | [165] |
| CsPbBr$_3$ QDs/G.O. | 100 W Xe, AM 1.5G filter | ethyl acetate | 357.40 | 99.1 | 58.70 | 29.6 | 1.58 | [165] |
| CsPbBr$_3$ | 100 W Xe, AM 1.5G filter | CO$_2$ and H$_2$O vapour | 33.42 | 100 | - | - | - | [166] |
| CsPbBr$_3$/ZIF-8 | 100 W Xe, AM 1.5G filter | CO$_2$ and H$_2$O vapour | 46.49 | 100 | - | - | - | [166] |
| CsPbBr$_3$/ZIF-67 | 100 W Xe, AM 1.5G filter | CO$_2$ and H$_2$O vapour | 88.89 | 100 | - | - | - | [166] |
| 15%-CsPbBr$_3$ QDs/UiO-66 | 300 W Xe, 420 nm filter | H$_2$O/ethyl acetate | 222 | - | 98.57 | 3.08 | - | [167] |
| CsPbBr$_3$ NCs | 300 W Xe, 420 nm filter | ethyl acetate | 29.60 | 100 | 3.62 | 2.79 | 0 | [158] |
| CsPbBr3 NCs/Pd NS (100) | 300 W Xe, 420 nm filter | ethyl acetate | 41.382 | 97.6 | 7.92 | 3.07 | 0.50 | [158] |
| CsPbBr$_3$ NCs/Pd NS (300) | 300 W Xe, 420 nm filter | ethyl acetate | 59.08 | 96 | 12.63 | 10.41 | 1.17 | [158] |
| CsPbBr$_3$ NCs/Pd NS (600) | 300 W Xe, 420 nm filter | ethyl acetate | 101.39 | 93.5 | 5.77 | 5.26 | 3.29 | [158] |
| CsPbBr$_3$ NCs/Pd NS (900) | 300 W Xe, 420 nm filter | ethyl acetate | 52.66 | 94.7 | 3.90 | - | 1.40 | [158] |
| CsPbBr$_3$ NCs | 150 W Xe, AM 1.5G filter | ethyl acetate/IPA | 25.72 | 90.3 | 3.35 | 2.06 | 1.64 | [168] |
| CsPbBr$_3$ NCs/a-TiO$_2$ (10) | 150 W Xe, AM 1.5G filter | ethyl acetate/IPA | 106.56 | 90.5 | 7.73 | 10.12 | 5.08 | [168] |
| CsPbBr$_3$ NCs/a-TiO$_2$ (20) | 150 W Xe, AM 1.5G filter | ethyl acetate/IPA | 193.36 | 95.5 | 11.71 | 20.15 | 4.38 | [168] |
| CsPbBr$_3$ NCs/a-TiO$_2$ (30) | 150 W Xe, AM 1.5G filter | ethyl acetate/IPA | 140.09 | 93.7 | 8.05 | 14.4 | 4.42 | [168] |
| CsPbBr$_3$ NCs/a-TiO$_2$ (50) | 150 W Xe, AM 1.5G filter | ethyl acetate/IPA | 79.25 | 87.4 | 8.72 | 6.47 | 5.01 | [168] |

The instability of the commonly-used MHPs has been reported in the current literature as one of the main drawbacks of aqueous solutions [162]. Low-polar, non-aqueous solvents, such as ethyl acetate, have been widely selected in many studies in order to assure a longer stability of the particles during the photocatalytic processes [165,169]. In order to overcome the stability issues, $CO_2$ can be reduced to its gaseous form at high temperatures, however, the application of a solid–vapour system may be limited by several factors, including a lower efficiency and selectivity [170]. On the other hand, the separation and transfer of the photogenerated charges may also represent a limit to the photocatalytic applications [90]. The recorded high photoluminescent quantum yield can be interpreted as the fast consumption of the excited electron−hole pairs in the radiative charge recombination before the chemical reactions [171]. An efficient separation of the electron–hole pairs, temporally and spatially, can be easily resolved by coupling MHPs' interfaces in compositions with other materials, including $TiO_2$, graphene oxide or metal-organic frameworks [165–168]. As far as future prospects go, it seems that the composition of multiple and suitable semiconductors may lead to the potential acceleration of the photoinduced electron transfer, inducing greater catalytic reaction sites, and it can also facilitate the injection of the free electrons in the subsequent chemical reactions [172]. In conclusion, even if the employment of MHPs in $CO_2$ reduction photocatalysis is still at its first stage, previously reported results demonstrate their great potential for progress in the near future.

### 4.2. Photocatalytic $H_2$ Evolution

Heterogeneous photocatalysis based on solar-light energy is considered among the promising ways to generate hydrogen, as a renewable approach to facing the recent energy crisis and demand. As in the majority of light-driven applications, the well-known $TiO_2$-based photocatalysts have demonstrated great potential as a low-cost and environmentally friendly approach to solar-hydrogen production. However, the principal barriers are the rapid recombination of photo-generated charges, as well as backward reactions and low activation under visible light irradiation [16]. The optical and electrochemical properties of MHPs may provide a new paradigm for solar fuel production, and recently have also been tested as ideal candidates in photocatalytic hydrogen generation [84].

A summary of recent approaches to photocatalytic hydrogen evolution, by means of MHPs and their composites, has been summarized in Table 3.

**Table 3.** Comparison of $H_2$ generation efficiency of different hybrid organic–inorganic perovskites and their composites reported in the literature.

| Photocatalyst | Irradiation | Medium | HER µmol h$^{-1}$ g$^{-1}$ | Apparent Quantum Efficiency (%) | Ref. |
|---|---|---|---|---|---|
| MAPbBr$_3$ | λ ≥ 420 nm | mixed HI/HBr acid | 11.20 | - | [173] |
| MAPbBr$_3$/Pt | λ ≥ 420 nm | mixed HI/HBr acid | 33.60 | - | [173] |
| MAPbBr$_{3-x}$I$_x$ | λ ≥ 420 nm | mixed HI/HBr acid | 1021.20 | - | [173] |
| MAPbBr$_{3-x}$I$_x$/Pt | λ ≥ 420 nm | mixed HI/HBr acid | 2604.80 | 8.10 (450 nm) | [173] |
| MAPbBr$_3$ | λ ≥ 420 nm | HI acid | 11.31 | - | [53] |
| DMF-MAPbBr$_3$ | λ ≥ 420 nm | HI acid | 22.62 | - | [53] |
| DMSO-MAPbBr$_3$ | λ ≥ 420 nm | HI acid | 31.67 | - | [53] |
| Pt:DMSO-MAPbBr$_3$ | λ ≥ 420 nm | HI acid | 57.00 | - | [53] |
| MAPbBr$_3$ | 300 W Xe, λ ≥ 420 nm | HI solution | 14.00 | - | [174] |
| MAPbBr$_3$/Pt | 300 W Xe, λ ≥ 420 nm | HI solution | 40.00 | - | [174] |
| MAPbBr$_3$/rGO | 300 W Xe, λ ≥ 420 nm | HI solution | 938.90 | 1.4 (450 nm) | [174] |
| MAPbBr$_3$ | λ ≥ 420 nm | saturated HI solution | 38.00 | - | [162] |
| Pt/MAPbBr$_3$ | λ ≥ 420 nm | saturated HI solution | 90.00 | - | [162] |
| TiO$_2$-Pt/MAPbBr$_3$ | λ ≥ 420 nm | saturated HI solution | 620.00 | - | [162] |
| Pt/TiO$_2$-MAPbBr$_3$ | λ ≥ 420 nm | saturated HI solution | 1986.67 | 70 (420 nm) | [162] |

The encapsulation technique was first applied in order to overcome the instability problems in hydrogen generation based on MHPs. In 2016, Crespo-Quesada et al. [175] presented their work on CH$_3$NH$_3$PbI-based photocathodes used as a light absorber and an efficient charge transfer under solar simulator irradiation. In their work, a simple encapsulation approach was used, leading to the generation of H$_2$ over 2 h in a mildly neutral electrolyte. Similarly, Nam et al. [176] applied a lift-off process to encapsulate the MHPs-based photoelectrode, achieving more precise desired morphological dimensions, and the possibility of additional layers with specific protective and photocatalytic functions. The prepared device showed remarkable results toward water oxidation as well as an excellent long-term stability in a strongly oxidizing electrolyte. The stability problem in the water splitting application is associated with the decomposition of the MHPs that usually occurs because of the ingress of water molecules through the pinholes. Thus, as an alternative, a remarkable enhancement in the stability was achieved by employing pinhole-free hole transport layers [177]. Another method was identified by Park and their group, who prepared a suitable solar-driven hydrogen evolution system based on bare MHPs using hydrohalic acids as medium [53]. In their work, photocatalytic H$_2$ generation was performed without co-catalysts or oxidants, and stable conditions were maintained over 160 h due to the dynamic equilibrium between phases. Lately, similar techniques have been employed using mixed aqueous HBr/HI solution. Powder samples of MHPs loaded with Pt co-catalyst particles, applied on the surface, demonstrated superb photocatalytic H$_2$ evolution activity, especially due to the efficient band gap and the subsequent separation at the interface of the materials [173]. The enhanced results of such a structure should be considered in future, with regards to other MHPs-based materials. In order to overcome the unfavorable and expensive approaches previously mentioned, Kim et al. investigated a simpler strategy for the utilization of conventional MHPs photocatalysts by designing a semi-transparent photocathode compatible with strongly acidic aqueous electrolytes [178].

Despite the potential results of the previously cited works, all-inorganic halide perovskites should be considered as well, as promising materials. These materials exhibited different advantages in comparison to other hybrid MHPs systems, including higher photocurrent density, higher photovoltage potential, and, more importantly, higher stability in contact with several acidic electrolytes [178].

## 4.3. Photocatalytic Removal of Organic Contaminants

Environmental industrial emissions of organic contaminants, due to their toxicity, are globally considered as a serious threat to human health and ecosystems. Recently, the impressive response achieved through MHPs has inspired applications well beyond those previously mentioned. In addition to the various well-known applications, MHPs have been utilized as remarkably efficient photocatalysts for numerous light-driven reactions, including wastewater treatments [179,180]. The first results in photocatalytic wastewater treatment belong to Aamir et al. [181], who reported the preparation of new hydroxyl ammonium lead halide perovskites, OHNH$_3$PbI$_2$Cl and OHNH$_3$PbCl$_3$, which showed suitable stability in an aqueous medium and visible light activity in the photocatalytic degradation of yellow dye under ambient conditions. Lead-based MHPs emerged as visible light photocatalysts, applied in the degradation of other organic dye contaminants such as rhodamine B (RhB). The efficiency was found to be enhanced in the presence of assistant oxidants, such as H$_2$O$_2$, and the evidence of reactive oxygen species production was defined [182]. Lead-free hybrid MHPs have been prepared by replacing lead with cadmium, a non-toxic 6p-block element, leading to higher stability, increased charge-carrier lifetimes, and lower charge carriers recombination rates [183]. Bismuth-based hybrid organic–inorganic MHPs have been prepared and tested as new effective visible light-responsive photocatalysts, for the degradation of different environment pollutants [101].

However, despite the recent success of organometallic halide perovskites, other analogue MHPs have been developed with competitive performances. Among them, the preparation of all-inorganic MHPs has recently gain a lot of attention, because they demonstrated better stability and the ability to absorb a wider range of the visible light region of the solar spectrum [90]. After the pioneering work of Protesescu et al. properties [77], the preparation of all-inorganic MHPs has been the focus of

much effort. Gao et al. [23] reported MHPs' potential photoactivity on organic dye degradation in 2017. In his work, all-inorganic $CsPbX_3$ (X = Cl, Br, I) quantum dots were promoted to study the potential effects of preparing various types of halide perovskites for the photocatalytic degradation of methyl orange (MO). Lately, similar evidence was obtained concerning $CsPbBr_3$ for the photodegradation of 2-mercaptobenzothiazole, an important heterocyclic aromatic compound, poorly biodegradable, toxic, and probably carcinogenic to humans [184]. In 2019, inorganic $CsPbBr_3$ quantum dots were successfully employed even for the removal of antibiotic residues. The photocatalytic activity was evaluated on the photodegradation of tetracycline hydrochloride in ethanol as a model reactant, showing its benefits as a novel visible light photocatalyst for wastewater applications [180].

Schünemann et al. [185] suggested a polystyrene-based colloidal crystal templating approach to synthesize macro/mesoporous $CsPbBr_3$. The as-prepared MHP, with inverse opal morphology, resulted in superior photocatalytic activity compared to the bare CsPbBr3 material. The results, potentially beneficial for other photocatalytic reactions, greatly enriched the MHPs-based novel research area.

Environmentally friendly alternative approaches, with respect to the commonly employed lead-based MHPs, have been successfully proposed by Reyes-Pérez et al. [179], by employing $CsSnBr_3$ in crystal violet photodegradation under visible light irradiation. A similar approach was employed for the degradation of methylene blue and methyl orange by nanostructures of $TlCdI_3$, synthesized by a new, efficient and quick mechanical method [186].

Lim et al. [187] have recently proposed a facile method for eight binary and six ternary perovskite-like and perovskite-derivative nanostructures, leading to potential applications in many hundreds of other similar materials. The optical and photocatalytic properties of perovskite-based nanomaterials were evaluated, and distinguished applications have been proposed.

Compositional engineering applied to MHPs allowed the great tunability of the band gaps over a wider range of the solar-light spectrum, making the photons effectively harvest sunlight. The improved stability and successful alcohol-based photocatalytic systems based on MHPs, have been developed by employing a lead-free double material, $Cs_2AgBiBr_6$, used for the photocatalytic degradation of four types of ionic dyes. Among the dyes degraded, rodamine B was found to be almost completely degraded, up to 98%, within two hours of irradiation [188].

Inspired by the remarkable efficiency of MHPs, heterostructured and composite perovskite-based photocatalysts have also been prepared. In 2017, Pu et al. [189] studied the degradation of p-nitrophenol under visible light irradiation, by preparing methylamine lead bromide perovskite/protonated graphitic carbon nitride nanocomposites. In the study, a higher stability and charge transfer ability were recorded; in particular, the photoexcited electron on the hybrid perovskite conduction band was found to preferentially transfer to the carbon-base sheets, leading to an enhanced charge separation. Their work demonstrated the possibility of modulating the interfacial charge transfer behavior, achieving a superior charge separation efficiency in heterostructured photocatalyst. The band alignment in the formamidinium lead bromide-based perovskite and $TiO_2$ has been composited, in order to demonstrate the enhancement in the selective photocatalytic oxidization of benzylic alcohols by funneling the photogenerated electrons from the perovskite into the titanium dioxide [172].

Following these principles, all-inorganic perovskites have also been applied in composites. Schünemann et al. [190] synthesized a $CsPbBr_3/TiO_2$ composite via low-temperature wet-impregnation, and demonstrated its enhanced visible light activity towards the selective oxidation of benzyl alcohol. ESR (electron spin resonance) analyses further showed the transfer of photoexcited electrons within the $CsPbBr_3$ to the $TiO_2$, and the generation of superoxide radicals. Similar results have been achieved by anchoring $Cs_3Bi_2I_9$ on UV100–$TiO_2$ nanoparticles used as visible light photocatalyst for methanol oxidation in formaldehyde, with a favorable decrease in the recombination of photoinduced charge carriers, and a suitable band gap for visible light driven photocatalytic applications [191]. Novel strategies for providing water resistance and increasing the photocatalytic efficiency of halide perovskites in aqueous medium, through unabated charge extraction, have been recently exploited

in effective composite systems, which help to extract the photo-generated charge carriers, as well as providing water resistance. In Type II and Type I or Quasi Type II systems based on $Cs_3Bi_2X_9$, oxalic acid and $TiO_2$, or $Ag_2S$, allowed the excitons to be generated by the back-illumination of the perovskite material [192]. The presence of the $TiO_2$ and the $Ag_2S$ stabilized the interactions with water molecules, and simultaneously efficiently decreased the photogenerated charges recombination.

Some of the results previously discussed are reported in Table 4 as a comparison of the main organic pollutants treated in the previous literature.

**Table 4.** Comparison of organic contaminants degradation percentage for different metal halide perovskites and their composites, reported in the literature.

| Photocatalyst | Band Gap (eV) | Irradiation | Pollutant | Degradation % | Ref. |
|---|---|---|---|---|---|
| $TlCdI_3$ | 2.9 | UV Light | methyl orange dye | 27 | [186] |
| $TlCdI_3$ | 2.9 | UV Light | methylene blue dye | 100 | [186] |
| $CsSnBr_3$ | 1.74 | Visible light | crystal violet dye | 73.1 | [193] |
| $CsSnBr_3$ | 2.4 | 500 W Xe, Visible light | methyl orange dye | 90 | [167] |
| $CsPbCl_3$ | 2.4 | 500 W Xe, Visible light | methyl orange dye | 82 | [167] |
| $CsPbBr_3$ | 2.26 | 300 W Xe, $\lambda \geq 420$ nm | tetracycline hydrochloride in ethanol | 76 | [180] |
| $CsPbBr_3$ | 2.26 | 300 W Xe, $\lambda \geq 420$ nm | tetracycline hydrochloride in water | 18 | [180] |
| $CsPbBr_3$ | 2.26 | 300 W Xe, $\lambda \geq 420$ nm | tetracycline hydrochloride in isopropanol | 53 | [180] |
| $CsPbBr_3$ | 2.26 | 300 W Xe, $\lambda \geq 420$ nm | methyl orange dye in ethanol | 70 | [180] |
| $OHNH_3PbI_2Cl$ | 3.7 | Solar light | dye Direct Yellow 27 | 93.98 | [194] |
| $OHNH_3PbCl_3$ | 3.9 | Solar light | dye Direct Yellow 27 | 82.19 | [194] |
| $Cs_3Bi_2I_9$-OA | 2.12 | Visible light | methylene blue in water | 62.1 | [192] |
| $Cs_3Bi_2I_9$-OA/$Ag_2S$ | - | Visible light | methylene blue in water | 88.8 | [192] |
| $Cs_3Bi_2I_9$-OA/$TiO_2$ | - | Visible light | methylene blue in water | 83.5 | [192] |
| $Cs_3Bi_2Br_9$-OA | 2.65 | Visible light | methylene blue in water | 26.6 | [192] |
| $Cs_3Bi_2Br_9$-OA/$Ag_2S$ | - | Visible light | methylene blue in water | 40.13 | [192] |
| $Cs_3Bi_2Br_9$-OA/$TiO_2$ | - | Visible light | methylene blue in water | 27.6 | [192] |
| $Cs_3Bi_2Br_9$ | 2.65 | Visible light | methylene blue in isopropanol | 66.3 | [192] |
| $Cs_3Bi_2Br_9$-OA NCs | 2.65 | Visible light | methylene blue in isopropanol | 58.8 | [192] |
| IO-$CsPbBr_3$ | 2.3 | simulated solar light, AM 1.5 | rhodamine 6G | 95 | [193] |
| Bulk-$CsPbBr_3$ | - | simulated solar light, AM 1.5 | rhodamine 6G | 75 | [193] |

### 4.4. Photocatalytic Organic Synthesis

The economic feasibility and the efficiency of photocatalytic processes have been considered, and they have also been identified as desirable approaches in the field of organic chemistry. Advances in the artificial formation of organic compounds have been achieved via photoredox catalysis with semiconductors of different nature [195–197]. The traditional photocatalysts employed have shown limits, such as difficult preparations, the uneconomic content of high-cost noble metals, instability, and in certain cases, low activities, leading to the necessity of new, easy-to-produce, economical and effective alternatives. Given the widespread success of MHPs in many light-driven applications, recently many researchers have questioned the possibility of their use in highly efficient photocatalytic organic molecules synthesis. Some of the mentioned research is summarized in Table 5.

**Table 5.** Comparison in different organic synthesis reactions of different metal halide perovskites and their composites, reported in the literature.

| Photocatalyst | Irradiation | Photocatalytic Reaction | Conversion % | Selectivity % | ref. |
|---|---|---|---|---|---|
| $TiO_2$ | simulated light irradiation, AM 1.5G | oxidation of benzylic alcohol | 15 | 95 | [172] |
| $FAPbBr_3$ | simulated light irradiation, AM 1.5G | oxidation of benzylic alcohol | 15 | 99 | [172] |
| nano-$FAPbBr_3$ | simulated light irradiation, AM 1.5G | oxidation of benzylic alcohol | 11 | 99 | [172] |
| 15% $FAPbBr_3/TiO_2$ | simulated light irradiation, AM 1.5G | oxidation of benzylic alcohol | 63 | 99 | [172] |
| 15% $FAPbBr_3/SiO_2$ | simulated light irradiation, AM 1.5G | oxidation of benzylic alcohol | 13 | 99 | [172] |
| 15% $FAPbBr_3/TiO_2$-M | simulated light irradiation, AM 1.5G | oxidation of benzylic alcohol | 37 | 99 | [172] |
| 15% $FAPbBr_3/TiO_2$ | $\lambda \geq 500$ nm | oxidation of benzylic alcohol | 13 | 99 | [172] |
| 15% $FAPbBr_3/TiO_2$ | without light irradiation | oxidation of benzylic alcohol | 0 | 0 | [172] |
| $CsPbI_3$ | Vis LED, 420–700 nm | thiophenol coupled to disulfide | 58 | - | [198] |
| $CsPbBr_3$ | Vis LED, 420–700 nm | thiophenol coupled to disulfide | 98 | - | [198] |
| $CsPbBr_2Cl$ | Vis LED, 420–700 nm | thiophenol coupled to disulfide | 98 | - | [198] |
| $CsPbBr_{0.5}Cl_{0.5}$ | Vis LED, 420–700 nm | thiophenol coupled to disulfide | 68 | - | [198] |
| $CsPbBrCl_2$ | Vis LED, 420–700 nm | thiophenol coupled to disulfide | 35 | - | [198] |
| $CsPbCl_3$ | Vis LED, 420–700 nm | thiophenol coupled to disulfide | 12 | - | [198] |
| $CsPbCl_3$ | Vis LED, 420–700 nm | thiophenol coupled to disulfide | 93 | - | [198] |
| $CsPbCl_3 + Br_2$ | Vis LED, 420–700 nm | thiophenol coupled to disulfide | 62 | - | [198] |
| $Cs_3Bi_2Br_9$ | visible light $\geq 420$ nm | alcoholysis of styrene oxide in IPA (isopropanol) | >99 | - | [195] |
| $Cs_3Bi_2Br_9$ | visible light $\geq 420$ nm | alcoholysis of styrene oxide in IPA | >99 | - | [195] |
| $CsPbBr_3$ | visible light $\geq 420$ nm | alcoholysis of styrene oxide in IPA | 1 | - | [195] |
| $CsPbI_3$ | visible light $\geq 495$ nm | polymerization of 3,4-ethylenedioxythiophene | 32.6 | - | [199] |
| $CsPbBr_3$ | Blue LED 455 nm | $\alpha$-alkylation of aldehydes | >99 | 96 | [200] |
| $CsPbBr_3$ NCs | 12 W Blue LED, 455 nm | synthesis aldehyde | 85 | - | [19] |
| $CsPbBr_3$ NCs | 12 W Blue LED, 455 nm | synthesis aldehyde | 52 | - | [19] |
| $CsPbBr_3$ NCs | 12 W Blue LED, 455 nm | synthesis tertiary amines | 90 | - | [19] |

**Table 5.** *Cont.*

| Photocatalyst | Irradiation | Photocatalytic Reaction | Conversion % | Selectivity % | ref. |
|---|---|---|---|---|---|
| CsPbBr3 NCs | 12 W Blue LED, 455 nm | synthesis tertiary amines | 79 | - | [19] |
| CsPbBr$_3$ NCs | 12 W Blue LED, 455 nm | cyclization of benzaldehyde phenylhydrazone | 88 | - | [19] |
| MAPbBr$_3$ | 12 W Blue LED, 455 nm | cyclization of benzaldehyde phenylhydrazone | 75 | - | [19] |
| CsPbBr$_3$ NCs | 12 W Blue LED, 455 nm | cyclization of ethyl (Z)-3-phenyl-3-(phenylamino)acrylate | 93 | - | [19] |
| MAPbBr$_3$ | 12 W Blue LED, 455 nm | cyclization of ethyl (Z)-3-phenyl-3-(phenylamino)acrylate | 65 | - | [19] |
| CsPbBr$_3$ NCs | 12 W Blue LED, 455 nm | coupling of benzoic acid with 4-bromotrifluorobenzene | 78 | - | [19] |
| CsPbBr$_3$ | 4.6 W Blue LED | photopolymerized styrene | 12 | - | [201] |
| no perovskite | 4.6 W Blue LED | photopolymerized styrene | 3.2 | - | [201] |
| no light | 4.6 W Blue LED | photopolymerized styrene | 1 | - | [201] |

Huang et al. [172] studied the efficiency in photocatalytic oxidization of benzylic alcohols, by utilizing an energy band alignment within formamidinium lead bromide MHPs and TiO$_2$ irradiated by solar-light. In their work, the authors highlighted the necessity of a capping agent in improving the instability of the MHPs-based composites. As alternative, in 2017, all-inorganic CsPbI$_3$ quantum dots were applied for the synthesis of conducting polymers, which are widely used, especially in optoelectronic devices, and are usually prepared by slow and difficult chemical or electrochemical methods [199]. Recently, Zhu et al. [200] compared two lead-based MHPs colloids, for selective carbon−carbon bonds for the formation of aldehydes, leading to the development of many other possibilities for potential organic compounds synthesis. In fact, the same catalytic systems shown were evaluated for the photocatalytic formation of symmetrical and unsymmetrical disulfides from thiol [198]. Moreover, Tan et al. [201] tested the stability of lead-based MHPs protected by high molecular weight polymers fully immersed in water, further demonstrating the success of these materials in photoactivated organic synthesis. A three-component hybrid perovskite-based solar photocatalyst cell (NiO$_x$/FAPbBr$_3$/TiO$_2$) was adopted for photocatalytic organic synthesis, due to its capacity for bond-activation with high selectivity, and high conversion rates under ambient conditions [202].

Among the possible lead-free and more eco-friendly alternatives, Bi-based MHPs have been shown to be promising, with advantageous low toxicity and higher air-stability [203]. As an example, lead-free Cs$_3$Bi$_2$Br$_9$ has recently demonstrated exceptional high activity and selectivity in solar light-driven photocatalytic organic synthesis [195]. The reasons for such an unexpected catalytic ability are attributed to the combination of the photocatalytic process and the presence of proper Lewis acidic centers on the surface of the photocatalyst, leading to a higher activity with respect to the lead-based counterpart.

The development of MHPs in photocatalytic organic chemistry, even if at its early stage, has shown promising results, offering a preliminary framework for future investigation.

## 5. Current Limitations and Future Perspectives

The core of our work is to review the recent achievements in MHPs-based materials applied in photocatalytic processes. Recently, the discovery of their easy preparation and processing, coupled with remarkable optical and electrochemical properties, has attracted the interest of several academic fields. Therefore, MHPs have been considered in the new generation of materials with great potential in many light-driven applications, leading to an incredibly rapid development. The field of MHP-based photocatalysis is still in a relatively early stage of discovery, and some limits still need to be addressed in order to achieve further improvement and promote future industrial commercialization. Their feasibility is still hindered by relatively poor stability in particular environments, and in certain cases, the efficiency can be further improved. Structural and chemical stability, as well as temperature, humidity and long-term irradiation, represent the main gaps to be overcome in order to design and optimize MHPs-based photocatalysts.

In recent years, several approaches have been developed in order to achieve promising and stable photocatalytic reaction environments by means of MHPs [204–206].

Structural and chemical stabilities include the lack of polymorphism [207], and a lack in resistant crystalline phases [208]. These effects can be easily overcome by selecting suitable preparation conditions, such as temperature and pressure [209], or by modifying the MHPs morphological compositions [210]. Generally speaking, the MA$^+$ and FA$^+$ organic precursors, MA$^+$-based MHPs, showed higher photocatalytic performance and superior thermal stability, but similar humidity problems [211]. Hybrid organic–inorganic structures have strict requirements for ion size, and the structural symmetry stability is considerably decreased by a small lattice expansion or distortion. In general, it has been reported that the tolerance factor, mainly dependent on the effective radii of the employed ions [212], may influence the structural stability, and thus by replacing or mixing the compositions of MHPs with different sized ions is a possible means to obtain crystal structures with greater stabilities [213]. Furthermore, thermal and humidity disadvantages can be faced by mixing

or replacing organic cations with inorganic elements, for example, cesium has emerged as the most widely used inorganic precursor [110,214,215].

More specifically, several methods have been tested in order to enhance environmental stability in relation to target photocatalytic reactions, including $CO_2$ reduction, $H_2$ evolution, organic matter photodecomposition or organic transformations. The relative HOIPs (hybrid organic inorganic perovskite) sensitivity to degradation by polar molecules can be addressed by using low-polarity solvents for $CO_2$ reduction, such as ethyl acetate or acetonitrile, creating stable photocatalytic reaction conditions [164]. Further enhanced stability can be achieved by combining HOIPs with graphene or graphitic carbon nitrile, by passivation through the interaction between the surface groups [174]. In addition, the compositions with $TiO_2$ or transition MXenes structures can help to speed up the charge separation, by acting as reaction sites to improve the overall stability [216]. Morphology modification, and substitution of halogens in the HOIPs or the ions in the A- and B-sites' structure, have also been proven as suitable approaches to enhancing the stability toward decomposition [163,164,167,192]. In the same way, low-polarity solvents may be employed in photocatalytic organic synthesis, including non-polars such as toluene, dichloromethane, tetrahydrofuran, hexane etc. [19,166,195,200,217].

Suitable compositions, and re-engineering of the HOIPs-based systems, can be useful approaches not only to achieving higher transformation efficiency, but also to stabilizing the photocatalysts in photo-oxidation processes [172]. Besides the synthesis of organics into value-added products, the complete mineralization of organic pollutants of MHPs photocatalysts has also been investigated in toluene [218].

The precipitation–solubility equilibrium, between the HOIPs phase and the soluble ionic species, represents one among the creative approaches to facing the stability problem. For example, saturated hydrogen iodide (HI) acid aqueous solutions using HOIPs polycrystalline powders were found to be a suitable method for photocatalytic $H_2$ production [53]. Due to the dynamic equilibrium between the MHPs powders and the ionic species in the saturated solution, the material was found to remain stable for several hours under continuous irradiation. Similar principles have been employed by using HBr/HI mixed acid solution [173]. Improved stability and enhanced hydrogen photocatalytic activity were achieved by composing HOIPs in supports, such as $TiO_2$ [162], rGO [219] and black phosphorus [220].

As an alternative, a straightforward stabilization method consists of encapsulating the HOIPs by means of a protective layer. The core-shell particles may prevent direct contact between the material and the destabilizing environment. This approach was used for $CO_2$ reduction [221], as well as in organic pollutants photodecomposition [222]. The core-shell structure achieved the valuable stabilization of the HOIPs, not only in low-polarity solvents, but also in a high-polarity solvent, like water. Adding hydrophobic ligands to coat the surface is another waterproof effective method [223].

In photoelectrochemical cells, used in $H_2$ evolution and $CO_2$ reduction reactions, HOIPs have been coated with hole and electron transport layers (HTL and ETL, respectively), enhancing the overall device stability. Two main options have been exploited: the implementation of MHPs photoelectrodes in separated compartments [54], and the HOIPs integrated photoelectrode [175]. The mentioned approaches offered relatively successful solutions. However, the development of MHPs that are intrinsically stable under common conditions represents an ideal scenario for photocatalytic reactions. In general, replacing or partially replacing the organic fraction with all-inorganic MHPs has resulted in higher stability, due to the enhanced formation energy, lower configurational entropy, and the non-volatile and less hygroscopic nature [224]. Many approaches have been developed in different photocatalytic reactions, including $CO_2$ reduction [158,163–165,168] and organic dye degradation [167,180,186,192–194].

Lead-based MHPs have shown very promising performances in photocatalytic processes, but due to the high and well-known toxicity of Pb, for both the ecosystem and environment, it should be replaced with more eco-friend elements. Generally, germanium- and tin-based perovskite systems showed rapid oxidation and uncontrolled crystallization, resulting in poorly efficiency. Bismu- and

antimony-based MHPs showed promising results, with stability comparable to lead-based MHPs [85]. However, appropriate efficiencies, due to the unsuitable band gaps, have not reached the traditional standards. Thus, a better understanding of photogenerated carrier dynamics at the interfaces needs to be addressed, in order to effectively improve the performance and stability of all lead-free MHPs.

There is a lack of total understanding of the optical and chemical reactions, and microscopic physics mechanisms, in the majority of the MHPs' applications. Therefore, the development of complete theoretical models is necessary to explain the remarkable efficiency of these materials when applied in photocatalytic processes. Theoretical research can improve the understanding of the established activities of MHPs, and they may provide strategies to develop simpler materials and structures with even higher efficiencies.

Thus, MHPs-based photocatalysts currently need further developments in order to be effective with regards to the energy and environmental needs of our society. However, the state of the art has demonstrated that MHPs represent an emerging generation of materials with several applications, including in photocatalytic processes. Despite the number of remaining challenges, it is important to remind ourselves that the commercial successes of emerging materials and technologies, such MHPs and their applications, have never been achieved in just a few decades.

Our research intended to present a general overview of the main progresses in MHPs-based photocatalysts, recognizing their great potential as future sources of interdisciplinary and multidisciplinary research.

**Author Contributions:** Conceptualization, B.-M.B.; data curation, B.-M.B.; writing—original draft preparation, B.-M.B.; writing—review and editing, B.-M.B.; supervision, Y.P., D.W.B. All authors have read and agree to the published version of the manuscript.

**Funding:** This research was funded by Maa-ja vesitekniikan tuki Foundation.

**Conflicts of Interest:** The authors declare no conflict of interest.

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
