# Peer review of "Recent Progresses on Metal Halide Perovskite-Based Material as Potential Photocatalyst"

_catalysts, doi:10.3390/catal10060709_

Round 1
Reviewer 1 Report
Reviewer advises to consider the followings suggestions while preparing the revised version of the manuscript.
- Rather than highlighting TiO2 only, provide a concise and brief summary of the materials developments as a photocatalyst (e.g. metal based, metal-free etc.), and the emergence of perovskites as photocatalyst. If you could include a time-line of progress in materials development would be great. A latest review in this regard can be found here Chem. Soc. Rev., 2020,49, 1887-1931. I would stress to compare the perovskite with newly emerged but highly celebrated carbon nitride (for example, Phys. Rev. Mater. 2018, 2 (12), 125402; Appl. Phys. Lett. 112 (25), 253902) to depict the emergence of perovskite as a new material for photocatalytic applications.
- Include a separate section namely 'Perovskites as photocatalyst' and discuss the relevant physicochemical properties that qualify the perovskites as a photocatalyst for various applications.
- Perovskites are known for their instability in contact with moisture, light and air. Please include a section 'Stability of perovskites photocatalysts' to discuss the research strategies that have been adopted so far to stabilize the perovskites for meaningful photocatalytic applications. Particularly, for water-splitting, it is a great challenge for perovskite photocatalyst. Lesson can be learned from phosphorene (for example, J. Mater. Chem. A 6 (38), 18403-18408).
- Please include representative figures for relevant applications (i.e. water-splitting, CO2 reduction, purification etc.)
- The perspective section is largely full of vague comments. Please name a particular problem(for example, quantum yield, STH, stability, charge transport, electrodynamix etc.) with perovskites and recommend the useful strategies to overcome it. Be specific and precise.
- I didn't see R and O in the figure 1.
- Manuscript demands more care in preparation. it is irritating to see repeatedly the message (Error! Reference source not found.)
Author Response
REVIEWER 1
- Rather than highlighting TiO2 only, provide a concise and brief summary of the materials developments as a photocatalyst (e.g. metal based, metal-free etc.), and the emergence of perovskites as photocatalyst. If you could include a time-line of progress in materials development would be great. A latest review in this regard can be found here Chem. Soc. Rev., 2020,49, 1887-1931. I would stress to compare the perovskite with newly emerged but highly celebrated carbon nitride (for example, Phys. Rev. Mater. 2018, 2 (12), 125402; Appl. Phys. Lett. 112 (25), 253902) to depict the emergence of perovskite as a new material for photocatalytic applications.
Thank you for the suggestion, we add a table with some recently investigated materials for photocatalytic applications (Table 1).
- Include a separate section namely 'Perovskites as photocatalyst' and discuss the relevant physicochemical properties that qualify the perovskites as a photocatalyst for various applications.
Thank you for the suggestion, we provide a modification in the structure of the 3rd chapter, the changes have been highlighted.
- Perovskites are known for their instability in contact with moisture, light and air. Please include a section 'Stability of perovskites photocatalysts' to discuss the research strategies that have been adopted so far to stabilize the perovskites for meaningful photocatalytic applications. Particularly, for water-splitting, it is a great challenge for perovskite photocatalyst. Lesson can be learned from phosphorene (for example, J. Mater. Chem. A 6 (38), 18403-18408).
Thank you for the suggestion. I provided some information on the approaches currently used to enhance the environmental stability of MHPs. I found useful to present them according to the photocatalytic applications took in consideration. I discussed the possibility to provide capping agent or encapsulation of perovskite (ex. LINE 583, 623), or the modification in term of system or working conditions (ex. LINE 424, 465).
- Please include representative figures for relevant applications (i.e. water-splitting, CO2 reduction, purification etc.)
Thank you for the suggestion. I provide a modification in the manuscript (LINE 129-132) completed with Figure 1B.
- The perspective section is largely full of vague comments. Please name a particular problem (for example, quantum yield, STH, stability, charge transport, electrodynamix etc.) with perovskites and recommend the useful strategies to overcome it. Be specific and precise.
Thank you for the suggestion. In my point of you, due to the lack of deep knowledge in specific topic (QY, charge transport etc.) a deep discussion on metal halide perovskite in this direction still needs more experimental data not available or not coherent. Indeed, generally speaking of QY, also for many other photocatalytic materials, is always difficult because not many reports are available and sometimes they provide data not comparable. I provide a general presentation of limitation and perspective in the manuscript in order to be consistent with the reality of data available in literature. As example, I presented Low stability in ambient conditions and the different approaches use to face the problem including: capping agents, encapsulation or device-packaging method. About future perspective, I maintained the same approach; I reported the challenges that scientists are facing, as example the real comprehension of the reactions or the creation of mechanism models. In my opinion, many efforts are still needed to fully understand the potential of MHPs, especially for photocatalytic processes. On the other hand, this is review aims to be an insights on this scientific topic that has recently gain a remarkable interest for the reason explained in the manuscript. I would find confusing to discuss about more specific features, that are for sure necessary for an excellent photocatalyst, if in literature the data are missing or not coherent.
I will be available to further discuss this point. I do understand the concerns of the reviewer and I appreciated the comments.
- I didn't see R and O in the figure 1.
Thank you very much for the comment, I provided to correct.
- Manuscript demands more care in preparation. it is irritating to see repeatedly the message (Error! Reference source not found.)
Thank you for the comment, I think this was an editing problem, I now fixed it.

Reviewer 2 Report
The paper provides a description of thorough investigation concerning the progress on metal halide perovskite-based material as potential photocatalysts. The conclusions are quite interesting and the explanation and ideas presented in the paper may be very helpful for understanding the process in question. This quite important paper is publishable, therefore I recommend to accept this paper in present form.
Author Response
REVIEWER 2
The paper provides a description of thorough investigation concerning the progress on metal halide perovskite-based material as potential photocatalysts. The conclusions are quite interesting and the explanation and ideas presented in the paper may be very helpful for understanding the process in question. This quite important paper is publishable, therefore I recommend to accept this paper in present form.
Thank you very much for the comment and your revision.

Reviewer 3 Report
This is a comprehensive review on metal halide perovskite-based materials with a view to photocatalysis and contains a great amount of valuable information and references. The review is also well structures and contains large and neatly arranged tables which are very helpful in keeping overview. So all in all it is a useful review, however, some revision should be performed nonetheless before publication, as outlined below.
p. 1: The first paragraph (line 32 – 41) consists of banalities and should be omitted. Similarly, line 42 – 45 are platitudes and should be omitted.
p. 2, line 58 – 60: It is claimed that photocatalytic processes are applied “in particular to treat polluted air and water”. The authors should clearly present a table of these systems applied commercially on a large scale or otherwise delete this statement.
p. 2, line 62 – 65: The authors claim that “inspired by natural photosynthesis, the photocatalytic processes have been applied in different fields such as degradation of organic compound in liquid and gaseous phase, water splitting, carbon dioxide reduction and photocatalytic organic synthesis. First, degradation of organic compounds has nothing to do with natural photosynthesis, and such reactions can therefore not be inspired by natural photocatalysis. Second, the mechanisms in natural photocatalysis are far away from the mechanisms of water splitting and carbon dioxide in laboratory systems, and in fact the laboratory systems are hitherto rather primitive compared to natural photosynthesis. It is difficult to see an inspiration of natural photosynthesis in the mechanisms of those laboratory systems.
p. 3, line 110 - 112: It should be indicated where these pilot plants are located, which are the reactions in those pilot plants and how much solar energy they harvest per time unit.
p. 3, line 122, p. 4, line 144, p. 5, line 175, p. 9, line 413, p. 14, line 555, and p. 15, line 570: It is noted: “Error! Reference source not found.” I do not understand this remark.
p. 5, line 157 – 159: It is stated: “Solar-cells have shown great implementation by means of MHPs ... leading to results comparable to the commercial standards of more traditional materials.” In this case, it should be explained here why such solar cells are hardly found in commercially available solar cells (although the answer might become evident below).
p. 5, line 177 and Figure 3: in the text (line 177) Cs+ is indicated as cerium. Clearly, this is wrong as Cs designates cesium and not cerium, the latter would be Ce. So either Cs has to be replaced in line 177 and in Figure 3 by Ce, or cerium in line 177 has to be replaced by cesium.
p. 9, line 403 – 407: These are banalities which should be omitted. Even worse, in the following lines it is suggested that the carbon dioxide problem is going to be solved with the systems presented in the following. These systems are, however, completely inappropriate to reduce the carbon dioxide in the atmosphere noteworthy. I strongly recommend to discuss this topic a bit realistic.
p. 10, Table 1: This is a very informative table, however, the letters are a bit small and hence it is difficult to read. Would it be possible to print this table in landscape (i.e. to rotate the table by 90° and expand it laterally)?
p. 13, line 511 – 514: A meso-mesoporous photocatalyst is proposed for waste-water treatment. However, it completely unclear if this catalyst is effective at all under real conditions, i.e. a sewage plant, and which would be the advantage towards commercially used water cleaning such as ozone treatment which is highly efficient, cheap and well established.
p. 15, line 567: The authors emphasize the necessity for highly tolerant alternatives. However, it is not clear which property “highly tolerant” is related to.
Author Response
REVIEWER 3
This is a comprehensive review on metal halide perovskite-based materials with a view to photocatalysis and contains a great amount of valuable information and references. The review is also well structures and contains large and neatly arranged tables which are very helpful in keeping overview. So all in all it is a useful review, however, some revision should be performed nonetheless before publication, as outlined below.
- 1: The first paragraph (line 32 – 41) consists of banalities and should be omitted. Similarly, line 42 – 45 are platitudes and should be omitted.
Thank you for the comment. However, I have a different opinion. Semiconductor photocatalysis is one of the most promising tools to address energy crisis, global warming, and environmental pollution. My research is based on the research in new materials able to address these fundamental and current issues. I do understand that for scientists may be annoying to read the same introduction, but, in my opinion, it is important to introduce the reader to the topic, especially if the reader is not familiar with it. I appreciate the comment.
- 2, line 58 – 60: It is claimed that photocatalytic processes are applied “in particular to treat polluted air and water”. The authors should clearly present a table of these systems applied commercially on a large scale or otherwise delete this statement.
Thank you for the comment. I do understand that the table can be very useful also in the introduction, but, in my opinion, in the following chapter we provide enough proof of the use of photocatalysis in the field of pollutant treatments. I would not like to provide too many tables and confuse the reader on the main aim of the review. I appreciate the comment.
- 2, line 62 – 65: The authors claim that “inspired by natural photosynthesis, the photocatalytic processes have been applied in different fields such as degradation of organic compound in liquid and gaseous phase, water splitting, carbon dioxide reduction and photocatalytic organic synthesis. First, degradation of organic compounds has nothing to do with natural photosynthesis, and such reactions can therefore not be inspired by natural photocatalysis. Second, the mechanisms in natural photocatalysis are far away from the mechanisms of water splitting and carbon dioxide in laboratory systems, and in fact the laboratory systems are hitherto rather primitive compared to natural photosynthesis. It is difficult to see an inspiration of natural photosynthesis in the mechanisms of those laboratory systems.
Thank you for the comment. I will try to explain my point of view, but I am also agreed if the reviewer may find confusing the explanation for the future reader. I am open to discuss.
I am aware the differences, in terms of thermodynamic between photocatalytic processes, thermodynamically downhill (ΔG<0), are merely acceleration through the catalyst whereas photosynthetic processes, thermodynamically unfavorable (ΔG>0), require photochemical energy input to happen. The last decades have seen increasing research activity in photochemical processes for environmental and energy challenges. Light driven systems in excited states can promote an excitonic chemical conversion. In the literature, such systems are more commonly known as photocatalytic or artificial photosynthesis reactions. The similarity stands in the step of reaction pathway: ability to absorb light by species, ability to photogenerate and transfer charges, ability to perform chemical reaction starting from light energy sources.
I may considered to delete the comparison if not clear. I further thank for the comments and the chance to discuss.
- 3, line 110 - 112: It should be indicated where these pilot plants are located, which are the reactions in those pilot plants and how much solar energy they harvest per time unit.
Thank you for the comment. There are several pilot and semi-pilot plans, I update the references.
- 3, line 122, p. 4, line 144, p. 5, line 175, p. 9, line 413, p. 14, line 555, and p. 15, line 570: It is noted: “Error! Reference source not found.” I do not understand this remark.
Thank you for the comment, this was an editing problem, I fixed.
- 5, line 157 – 159: It is stated: “Solar-cells have shown great implementation by means of MHPs ... leading to results comparable to the commercial standards of more traditional materials.” In this case, it should be explained here why such solar cells are hardly found in commercially available solar cells (although the answer might become evident below).
Thank you for the comment; I completed the sentence that now (after modification) stands at LINE 168-170.
- 5, line 177 and Figure 3: in the text (line 177) Cs+ is indicated as cerium. Clearly, this is wrong as Cs designates cesium and not cerium, the latter would be Ce. So either Cs has to be replaced in line 177 and in Figure 3 by Ce, or cerium in line 177 has to be replaced by cesium.
Thank you for the comment, I provided to correct in the manuscript.
- 9, line 403 – 407: These are banalities which should be omitted. Even worse, in the following lines it is suggested that the carbon dioxide problem is going to be solved with the systems presented in the following. These systems are, however, completely inappropriate to reduce the carbon dioxide in the atmosphere noteworthy. I strongly recommend to discuss this topic a bit realistic.
Thank you for your point of view. You can now refer to modified LINE 414-417.
- 10, Table 1: This is a very informative table, however, the letters are a bit small and hence it is difficult to read. Would it be possible to print this table in landscape (i.e. to rotate the table by 90° and expand it laterally)?
Thank you very much for the suggestion, I think this can be done on the final edited document.
- 13, line 511 – 514: A meso-mesoporous photocatalyst is proposed for waste-water treatment. However, it completely unclear if this catalyst is effective at all under real conditions, i.e. a sewage plant, and which would be the advantage towards commercially used water cleaning such as ozone treatment which is highly efficient, cheap and well established.
Thank for the comment. I modified the sentence, now at LINE 518-521 in order to clarify the meaning.
- 15, line 567: The authors emphasize the necessity for highly tolerant alternatives. However, it is not clear which property “highly tolerant” is related to.
Thank for the comment. I modified the sentence, now at LINE 575-578.

Round 2
Reviewer 1 Report
Reviewer's queries are not addressed satisfactorily and to the point. Discernibly, authors didn't revise the manuscript in a great deal. Reviewer feels that authors need to work a lot to make this article a comprehensive review that would advance the field of perovskites based photocatalysts.
Author Response
REVIEWER 1
- Rather than highlighting TiO2 only, provide a concise and brief summary of the materials developments as a photocatalyst (e.g. metal based, metal-free etc.), and the emergence of perovskites as photocatalyst. If you could include a time-line of progress in materials development would be great. A latest review in this regard can be found here Chem. Soc. Rev., 2020,49, 1887-1931. I would stress to compare the perovskite with newly emerged but highly celebrated carbon nitride (for example, Phys. Rev. Mater. 2018, 2 (12), 125402; Appl. Phys. Lett. 112 (25), 253902) to depict the emergence of perovskite as a new material for photocatalytic applications.
Thank you for the suggestion. In the introduction we provide a table with the most traditiona photocatalyst used in the past years. We add a briefly time-line progress, with some of the principal references on the development of MHPs as material and photocatalysts.
- Include a separate section namely 'Perovskites as photocatalyst' and discuss the relevant physicochemical properties that qualify the perovskites as a photocatalyst for various applications.
Thank you for the suggestion. We modify the chapter 3.3 Outstanding properties of MHPs as photocatalyst dividing the discussion in sub-chapters by considering the main properties that make MHPs appealing as photoctalysts: Extended optical absorption range, Direct generation of free charge carriers, Long-range balanced and ambipolar charge carrier transport properties, Point-defect tolerance and reduced recombination centres, Point-defect tolerance and reduced recombination centres.
- Perovskites are known for their instability in contact with moisture, light and air. Please include a section 'Stability of perovskites photocatalysts' to discuss the research strategies that have been adopted so far to stabilize the perovskites for meaningful photocatalytic applications. Particularly, for water-splitting, it is a great challenge for perovskite photocatalyst. Lesson can be learned from phosphorene (for example, J. Mater. Chem. A 6 (38), 18403-18408).
Thank you for the suggestion. We provide a discussion in the 5 chapter, LINE 625-697, where we gave an overview of the most popular strategies adopted to enhance the stability in the different target photocatalytic reactions.
- Please include representative figures for relevant applications (i.e. water-splitting, CO2 reduction, purification etc.)
Thank you for the suggestion. We provide the representation of energy levels of the important photocatalytic reactions with respect to NHE at pH = 0 in Figure 2B.
- The perspective section is largely full of vague comments. Please name a particular problem (for example, quantum yield, STH, stability, charge transport, electrodynamix etc.) with perovskites and recommend the useful strategies to overcome it. Be specific and precise.
Thank you for the suggestion. We enhanced the discussion in perspective.
- I didn't see R and O in the figure 1.
Thank you very much for the comment, we provided to correct the error.
- Manuscript demands more care in preparation. it is irritating to see repeatedly the message (Error! Reference source not found.)
Thank you for the comment.
